# Antitumor effects of chloroquine/ hydroxychloroquine mediated by inhibition of the NF-κB signaling pathway through abrogation of autophagic p47 degradation in adult T-cell leukemia/lymphoma cells

Yanuar Rahmat Fauzi[1], Shingo Nakahata[1], Syahrul Chilmi[1], Tomonaga Ichikawa[1], Phawut Nueangphuet[2], Ryoji Yamaguchi[2], Tatsufumi Nakamura[3], Kazuya Shimoda[4], Kazuhiro Morishita[1]¤*

1 Division of Tumor and Cellular Biochemistry, Department of Medical Sciences, University of Miyazaki, Miyazaki City, Miyazaki Prefecture, Japan, 2 Department of Veterinary Medicine, Faculty of Agriculture, University of Miyazaki, Miyazaki City, Miyazaki Prefecture, Japan, 3 Department of Social Work, Faculty of Human and Social Studies, Nagasaki International University, Sasebo City, Nagasaki Prefecture, Japan, 4 Division of Hematology, Diabetes, and Endocrinology, Department of Internal Medicine, University of Miyazaki, Miyazaki City, Miyazaki Prefecture, Japan

¤ Current address: Project for Advanced Medical Research and Development, Project Research Division, Frontier Science Research Center, University of Miyazaki, Miyazaki, Japan
* kmorishi@med.miyazaki-u.ac.jp

## Abstract

Adult T-cell leukemia/lymphoma (ATLL) originates from human T-cell leukemia virus type 1 (HTLV-1) infection due to the activation of the nuclear factor-κB (NF-κB) signaling pathway to maintain proliferation and survival. An important mechanism of the activated NF-κB signaling pathway in ATLL is the activation of the macroautophagy (herafter referred to as autophagy in the remainder of this manuscript)-lysosomal degradation of p47 (NSFL1C), a negative regulator of the NF-κB pathway. Therefore, we considered the use of chloroquine (CQ) or hydroxychloroquine (HCQ) (CQ/HCQ) as an autophagy inhibitor to treat ATLL; these drugs were originally approved by the FDA as antimalarial drugs and have recently been used to treat autoimmune diseases, such as systemic lupus erythematosus (SLE). In this paper, we determined the therapeutic efficacy of CQ/HCQ, as NF-κB inhibitors, in ATLL mediated by blockade of p47 degradation. Administration of CQ/HCQ to ATLL cell lines and primary ATLL cells induced cell growth inhibition in a dose-dependent manner, and the majority of cells underwent apoptosis after CQ administration. As to the molecular mechanism, autophagy was inhibited in CQ-treated ATLL cells, and activation of the NF-κB pathway was suppressed with the restoration of the p47 level. When the antitumor effect of CQ/HCQ was examined using immunodeficient mice transplanted with ATLL cell lines, CQ/HCQ significantly suppressed tumor growth and improved the survival rate in the ATLL xenograft mouse model. Importantly, HCQ selectively induced ATLL cell death in the ATLL xenograft mouse model at the dose used to treat SLE. Taken together, our results suggest

**Data Availability Statement:** All relevant data are within the manuscript and its Supporting information files.

**Funding:** KM This work was supported in part by Grants-in-Aid for Scientific Research (B) (25293081 and 17H03581) (Kazuhiro Morishita) from the Japan Society for the Promotion of Science (JSPS) (https://www.jsps.go.jp/english/e-grants/)and by the Takeda Science Foundation (Kazuhiro Morishita) (https://www.takeda-sci.or.jp/). The funders had no role in study design, data collection and analysis, decision to publish, or preparation of the manuscript.

**Competing interests:** The authors have declared that no competing interests exist.

that the inhibition of autophagy by CQ/HCQ may become a novel and effective strategy for the treatment of ATLL.

## Introduction

Adult T-cell leukemia/lymphoma (ATLL) is an aggressive T-cell malignancy caused by infection with the retrovirus human T-cell leukemia virus type 1 (HTLV-1) [1]. HTLV-1 transmission requires cell-to-cell contact via a cell-containing body fluid through the following routes: mother-to-child transmission, sexual intercourse, and contaminated blood transfusion or contaminated needle puncture [2]. Currently, it is estimated that 15–20 million people are infected with HTLV-1 worldwide, but the exact number in the general population has remained unclear since most epidemiological studies are limited to endemic regions, such as Japan, Africa, Latin America, and the Caribbean islands [3, 4]. ATLL develops after a 40- to 60-year latency period in approximately 5% of HTLV-1 carriers [5, 6]. ATLL has been subdivided into four classifications: the acute, lymphoma, chronic, and smoldering types. The most common presentation is the acute type, which generally has a poor prognosis with a median survival time of approximately 4–6 months [6]. Although several new therapeutic approaches, such as anti-C-C chemokine receptor 4 (CCR4) antibodies and allogeneic hematopoietic stem cell transplantation (HSCT), have been established [7–9], the overall prognosis of aggressive-type ATLL patients remains unsatisfactory.

Previously, we reported that cell adhesion molecule 1 (CADM1), originally well known as a tumor suppressor in non-small cell lung cancer (NSCLC), is highly expressed in ATLL cells and promotes tumor growth and multiple organ invasion [10–12]. More recently, we discovered that an enhancer element influencing CADM1 expression in the CADM1 promoter region in ATLL cells contains a nuclear factor-κB (NF-κB)-binding sequence and that CADM1 expression is dependent on activation of the canonical NF-κB pathway [13]. The NF-κB pathway is well known for regulating innate immune cells and inflammatory T cells. NF-κB activation occurs via two major signaling pathways, the canonical and noncanonical pathways, which depend on activation of inhibitor of NF-κB α (IκBα) degradation and p100 processing, respectively [14]. In HTLV-1-infected T cells, the HTLV-1-encoded oncoprotein Tax stimulates the activation of both the canonical and noncanonical NF-κB pathways [15]. On the other hand, ATLL cells frequently lose Tax expression but acquire the ability to activate the canonical NF-κB pathway without Tax through genetic and epigenetic alterations involving the T-cell receptor/NF-κB signaling pathway, miR-31 silencing, and IL-17RB overexpression [16]. Along with persistent activation signals, the downregulation of the p97/NSFL1 cofactor p47, a negative regulator of the NF-κB pathway, was found to be essential for the constitutive activation of the NF-κB pathway in ATLL cells [17]. p47 is a major adaptor molecule of the cytosolic AAA ATPase p97 and was recently reported to bind to NEMO with Lys63-linked and linear polyubiquitin chains, leading to lysosome-dependent NEMO degradation and suppression of IκB kinase (IKK) activation. We recently identified that CADM1 overexpression is dependent on NF-κB activation through p47 downregulation by lysosomal-autophagic degradation in ATLL cells [13].

In this manuscript, we further analyzed the mechanism of autophagy activation in ATLL cells and determined whether autophagy suppression in ATLL cells is a potential therapeutic approach. Autophagy, roughly translated as "self-eating," is a lysosome-dependent multistep process that supports cell survival in a hypoxic, starvation, or stress environment. Autophagy

allows cells to recycle altered, damaged, or unused organelles and cellular components to prevent cancer development under normal circumstances. By contrast, during tumor progression, autophagy exerts protumoral activity by increasing resistance to anticancer drugs and providing nutrients for cancer cells [18]. Chloroquine (CQ) and hydroxychloroquine (HCQ), a less toxic metabolite of CQ, are widely used to treat malaria and inflammatory diseases, such as systemic lupus erythematosus (SLE) and rheumatoid arthritis (RA) [19]. Recently, it has been reported that CQ inhibits autophagy by impairing autophagosome-lysosome fusion [20].

In the present study, we investigated the anticancer effects of CQ and HCQ as monotherapies on ATLL cells *in vitro* and *in vivo*. ATLL cells exhibited high autophagic flux to maintain cell proliferation. CQ and HCQ were shown to induce apoptosis and inhibit ATLL cell growth both *in vitro* and *in vivo*. Autophagy inhibition by CQ or HCQ promoted p47 protein recovery along with loss of CADM1 expression and NF-κB pathway inhibition, leading to caspase-3-related apoptosis. Our findings suggest that inhibition of autophagy by CQ or HCQ may be a potential approach for ATLL treatment.

## Materials and methods

### Patient samples

Peripheral blood samples were collected at the Department of Medical Sciences, Faculty of Medicine, University of Miyazaki, in collaboration with the Miyazaki University Hospital. Informed consent was obtained from all the patients and the healthy donor. The Institutional Review Board approved this study at the Faculty of Medicine, University of Miyazaki, following the Declaration of Helsinki, the Ethical Guidelines for Medical and Health Research Involving Human Subjects, and the Ethics Guidelines for Human Genomic/Genetic Analysis Research. The diagnosis of ATLL was based on clinical features, hematological characteristics, and monoclonal integration of the HTLV-1 provirus according to Southern blot analysis results. PBMCs were isolated from the peripheral blood of ATLL patients and a healthy volunteer by Ficoll-Paque density gradient centrifugation (GE Healthcare Bioscience AB, Uppsala, Sweden) according to the manufacturer instructions. Primary cells were cultured in AIM-V medium (Thermo Fisher Scientific, Waltham, MA, USA) supplemented with 20% fetal bovine serum (FBS), 10 µM 2-mercaptoethanol (Thermo Fisher Scientific), and 0.75 µg/ml recombinant human IL2 (PeproTech, Rocky Hill, NJ, USA).

### Cell lines

Human interleukin-2 (IL-2)-independent ATLL cell lines (Su9T01, S1T, and MT1), HTLV-1-negative human T-cell acute lymphoblastic leukemia (T-ALL) cell lines (MOLT4 and JURKAT), and HTLV-1-infected T cell lines derived from HAM-TSP patients (HCT1, HCT4, and HCT5) [21, 22] were cultured in complete RPMI-1640 medium (Nacalai Tesque Inc, Kyoto, Japan) supplemented with 10% FBS and 50 µg/ml penicillin (10 U/ml)-streptomycin (10 µg/ml) (Fisher Scientific). The IL2-dependent ATLL cell lines KK1, KOB, and ST1 were cultured in complete RPMI-1640 medium supplemented with 0.75 µg/ml recombinant human IL-2. Both primary cells and cell lines were cultured at 37 °C in a humidified 5% $CO_2$ incubator.

### *In vivo* animal experiments

NOD/Shi-scid/IL-2Rγnull (NOG) mice were purchased from CLEA-Japan Inc. (Tokyo, Japan) and maintained under specific pathogen-free (SPF) conditions. Six- to eight-week-old male mice were xenografted subcutaneously (s.c.) in the right flank with 1 x $10^7$ MT2 cells or Su9T01 cells suspended in PBS in a final volume of 100 µl. Body weight and tumor volume

were measured every 3 days starting on the day of xenografting. Tumor volume was estimated using caliper measurement of the longest and shortest diameters of the tumor and then calculated using the following equation: shortest diameter$^2$ × longest diameter × 0.5. When the tumor volume reached 100–150 mm$^3$, mice were randomly divided into 2 (water or 50 mg/kg bw/day CQ, injected intraperitoneally (i.p.), n = 5 per group) or 3 groups (water, 6.5 mg/kg bw/day HCQ, or 60 mg/kg bw/day HCQ; per oral (p.o.), n = 7 per group), and treatment was started immediately. The mice were sacrificed after 21 days' daily treatment (CQ treated model) or seven days after 21 days' daily treatment (HCQ treated model). All mice were sacrificed with an inhalation overdose of isoflurane (FUJIFILM Wako Pure Chemical, Osaka, Japan).

For a survival model, NOG mice (minimum body weight: 22 grams) were intravenously injected with 1 x 10$^6$ Su9T01 cells suspended in PBS in a final volume of 100 μl. After 3 days, the mice were randomly divided into 2 groups (water or 60 mg/kg bw/day HCQ, p.o, n = 5 per group), and treatment was started immediately. HCQ or a control vehicle (water) was orally given for a 16-day period. Starting on the day of xenografting, body weight was measured every 3 days, and the general conditions were observed daily for a maximum period of 100 days [23]. In both the subcutaneous tumor model and survival model, humanitarian endpoints were based on the following criteria: decreased activity, decreased water intake, decreased appetite, hair growth, dirt on the coat, abnormal posture, or rapid weight loss (25% in 7 days). When the tumor was observed under the skin, the endpoint was a marked increase in tumor size (tumor diameter is 2 cm or 10% or more of body weight), and at that point, the mice were comforted by over-anesthesia for anesthesia. All animal experiments were approved by the Animal Experiment Review Board of the University of Miyazaki (2017-503-6).

## Results

### CQ and HCQ inhibit ATLL cell growth *in vitro*

To assess the effects of CQ or HCQ treatment on ATLL cell growth, four ATLL-derived cell lines (Su9T01, KK1, S1T, and ST1), two T-ALL cell lines (MOLT4 and JURKAT), and healthy human peripheral blood mononuclear cells (PBMCs) as a control were treated with different concentrations of CQ or HCQ (0, 1, 5, 10, 50, 100, and 200 μM) for 48 hours. Both CQ and HCQ were found to inhibit growth in all four ATLL cell lines in a dose-dependent manner (Fig 1A and S1A Fig). The IC50 values of CQ and HCQ were 40 +/- 10.2 and 25.9 +/- 15.1 μM, respectively (**S1 Table in** S1 File). Dose-dependent decreases in cell growth were also observed in primary ATLL cells from patients with acute-type ATLL after treatment with CQ or HCQ (IC50 = 19.5 +/-1.9 and 13.4 +/- 2.8 μM, respectively) (Fig 1B **and S2 Table in** S1 File). Interestingly, CQ or HCQ treatment also inhibited cell growth in HTLV-1-infected T cell lines derived from HTLV-1-associated myelopathy (HAM)/tropical spastic paraparesis (TSP) patients in a dose-dependent manner (IC50 CQ = 34.73 +/- 24.89 and IC50 HCQ = 30.37 +/- 12.46) (S1B Fig).

To determine the mechanism of apoptosis in CQ- or HCQ-treated ATLL cells, we evaluated the expression of cleaved caspase-3 as a marker of apoptosis by Western blot analysis. The results showed that the cleaved caspase-3 level was distinctly increased by CQ treatment in a time-dependent manner (Fig 1C). Moreover, an apoptosis assay using Annexin-V/DAPI double staining revealed a significant increase in apoptotic cells in both KK1 cells and S1T cells after 24 hours of exposure to CQ. Similar results were observed in HCQ-treated KK1 and S1T cells (Fig 1D). These results suggest that CQ and HCQ are potential drug candidates for ATLL treatment.

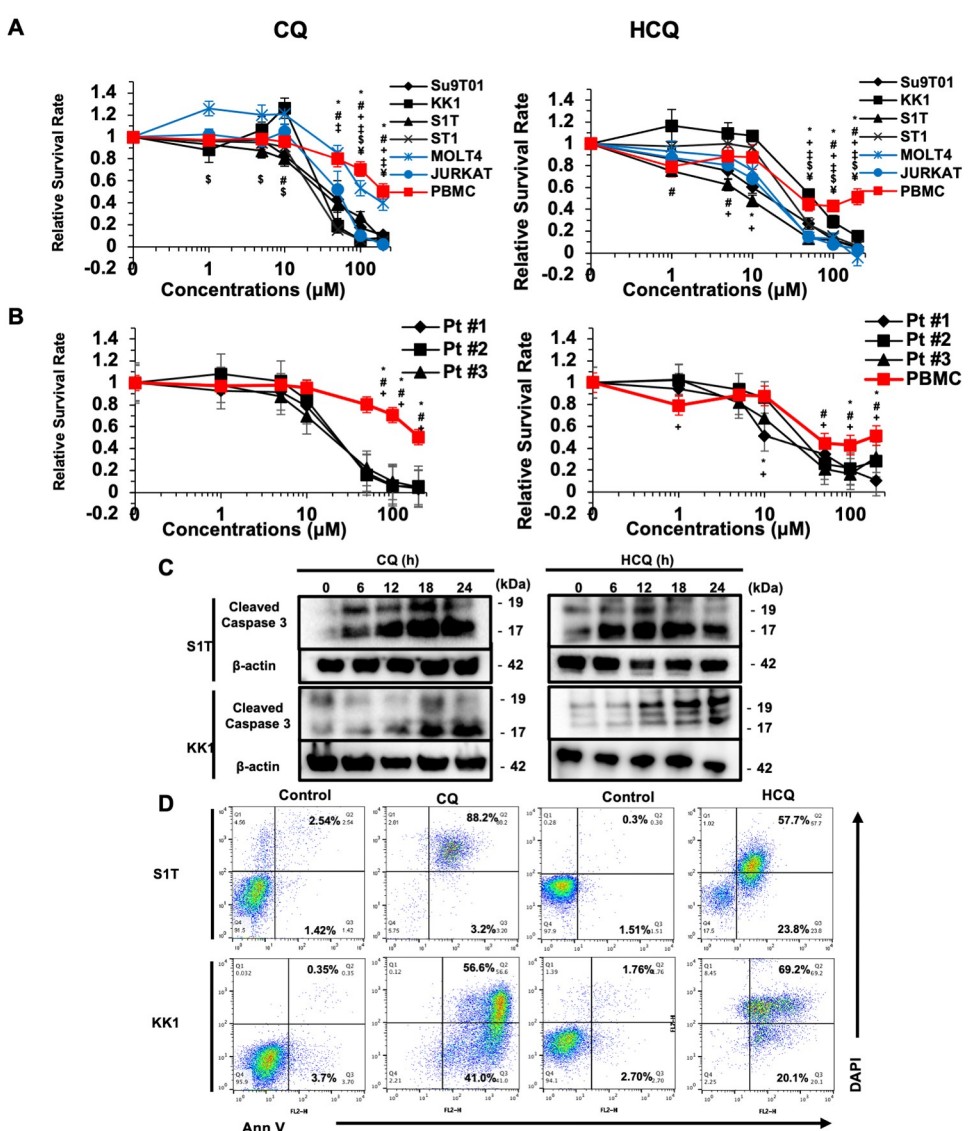

**Fig 1. CQ and HCQ inhibit ATLL cell growth *in vitro*.** (**A**) Four ATLL cell lines (Su9T01, KK1, S1T, and ST1), two T-ALL cell lines (MOLT4 and JURKAT), and PBMCs from a healthy donor were treated with varying concentrations of CQ or HCQ for 48 hours. Cell viability was determined by an MTT assay. Relative cell viability was calculated based on the percentage of untreated cells (0 μM). Data are presented as the mean ± SD (n = 3). *, #, +, ‡, $, or ¥ p < 0.05 for Su9T01, KK1, S1T, ST1, MOLT4, or JURKAT cells compared to healthy PBMCs, respectively. (**B**) Primary ATLL cells from four acute-type ATLL patients were treated with varying concentrations of CQ or HCQ for 48 hours. Cell viability was determined with the MTT assay. Relative cell viability was calculated based on the percentage of untreated cells (0 μM). Data are presented as the mean ± SD (n = 3). *, #, or + p < 0.05 for Pt #1, Pt #2, or Pt #3 compared to healthy PBMCs, respectively. (**C**) S1T and KK1 cell lines were treated with 50 μM CQ or 25 μM HCQ for 6, 12, 18, or 24 hours. The expression of cleaved caspase-3 was examined by Western blot analysis. β-actin was used as a loading control. (**D**) S1T and KK1 cell lines were treated with 50 μM CQ or 25 μM HCQ for 24 hours, double stained with Annexin-V and DAPI, and then analyzed by flow cytometry.

## CQ and HCQ inhibited autophagy and the canonical NF-κB pathway in ATLL cell lines

To determine whether CQ-mediated inhibition of ATLL cell growth is associated with suppression of NF-kB through blockade of p47 degradation via autophagic flux, we initially

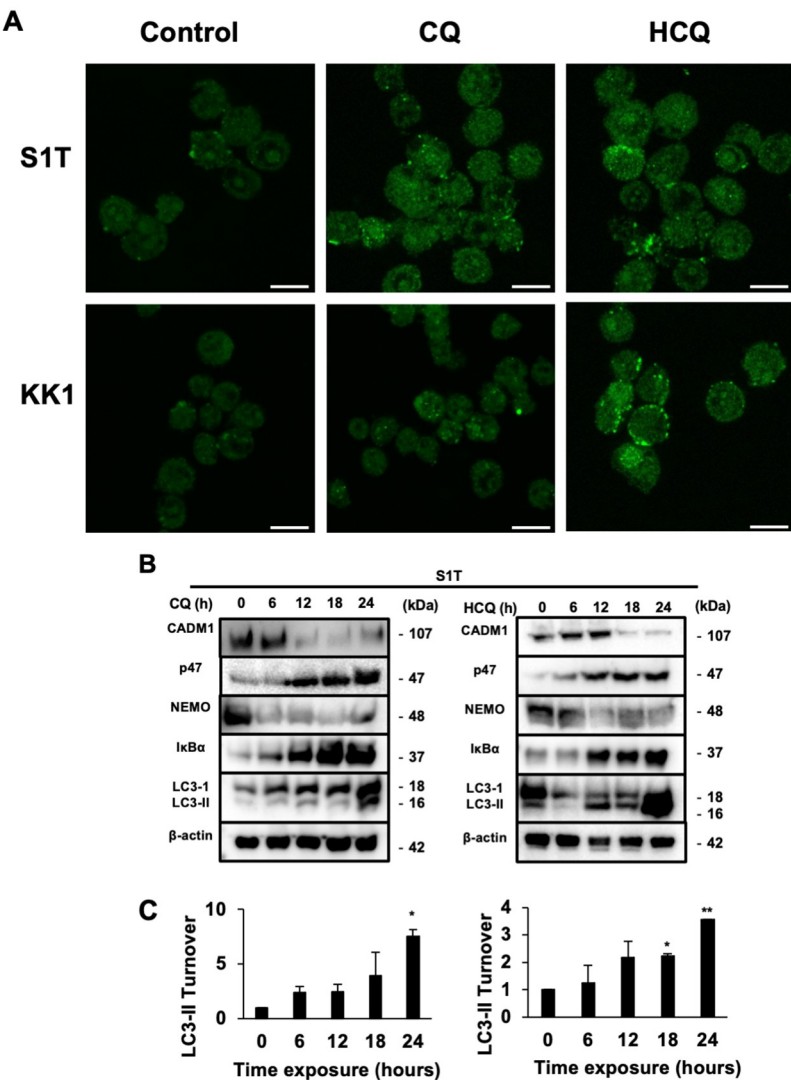

**Fig 2. CQ and HCQ inhibited autophagy and the canonical NF-κB pathway in ATLL cell lines.** (**A**) Representative confocal immunofluorescence of LC3 in KK1 and S1T cells after treatment with either 50 μM CQ or 25 μM HCQ for 24 hours. Scale bar 10μm. (**B**) Western blot analysis of CADM1, p47, and the indicated NF-κB and autophagic signaling proteins in S1T cell lines was performed after treatment with either 50 μM CQ or 25 μM HCQ for 6, 12, 18, or 24 hours. β-actin was used as a loading control. The cropped gels/blots are shown in the figure, and the full-length gels/blots are presented in S1 Raw images. (**C**) LC3-II turnover is presented as the mean ± SD (n = 2 independent experiments, representative blot was shown at Fig 2A). *p < 0.05, **p<0.01 compared to the control.

assayed the turnover of LC3-phospholipid conjugate (LC3-II) as an autophagic marker protein in ATLL cell lines (Su9T01, KOB, KK1, and S1T) that were either untreated or treated with various doses of CQ. CQ treatment caused the accumulation of autophagosomes, as indicated by the increased levels of LC3-II in all four ATLL cell lines, suggesting that CQ inhibits autophagic flux in ATLL cells (S2A Fig). Moreover, by immunofluorescence, we observed a significant accumulation of LC3 puncta in S1T and KK1 treated with CQ/HCQ compared to the untreated group (Fig 2A and S2B Fig).

We next determined whether inhibition of autophagy by CQ/HCQ treatment is associated with the accumulation of p47, resulting in the suppression of the activated NF-κB pathway, in

ATLL cells. Along with the accumulation of LC3-II, p47 protein levels increased after treatment of ATLL cell lines with CQ/HCQ (Fig 2B and 2C, S2C and S2D Fig). The upregulation of p47 induced increased NEMO degradation with accumulation of IκBα, resulting in reduced expression of CADM1 (Fig 2B and 2C, S2C and S2D Fig). Therefore, autophagy inhibition by CQ/HCQ treatment restores the expression of p47 and sequentially inhibits the activation of the NF-κB pathway, leading to apoptosis in ATLL cells.

## CQ and HCQ inhibit tumor growth and extend the lifespan of ATLL cells xenografted mice

To evaluate the *in vivo* therapeutic effect of CQ/HCQ on ATLL cells, we first established a subcutaneous xenograft model using immunodeficient NOD/Shi-scid/IL-2Rγnull (NOG) mice. Starting seven days after subcutaneous implantation of Su9T01/ATLL cells, CQ or a control vehicle (water) was intraperitoneally injected at a dose of 50 mg/kg bw daily, and tumor growth was monitored every 3 days for a 16-day period. CQ treatment significantly reduced the tumor volume of Su9T01 subcutaneous xenografts compared with control treatment (Fig 3A and S3A Fig). Then, HCQ was orally administered at a dose of 0, 6.5 or 60 mg/kg bw daily to NOG mice subcutaneously transplanted with Su9T01 or MT2 cells. HCQ treatment at 6.5 mg/kg bw or 60 mg/kg bw effectively inhibited tumor growth in Su9T01- and MT2-transplanted mice (Fig 3B and 3C, S3B and S3C Fig). H&E (Fig 3D) and immunohistochemical staining with an anti-cleaved caspase 3 antibody (Fig 3E) demonstrated that the number of apoptotic cells in tumor xenografts was significantly increased by treatment with HCQ (Fig 3F). The degeneration and necrosis of tumor cells can be seen in the HCQ treated groups (Fig 3D, arrows), which showed condensed hyperchromatic nuclei or fragmented nuclei with shrunk cytoplasm. Some broke down necrotic cells released the cellular debris to the extracellular area. The necrotic areas in the 6.5 mg/kg bw HCQ treated group were moderately observed in the tumor sheet and the center of the mass. The 60 mg/kg bw HCQ treated group showed hypereosinophilic with a large volume cytoplasm and highest intercellular spaces among groups (Fig 3D), along with predominantly observed necrotic area compared to 6.5 mg/kg bw HCQ group. In addition, mouse body weights were unchanged by CQ or HCQ treatment (S4 Fig). Finally, HCQ was administered orally at a dose of 0 or 60 mg/kg bw to NOG mice transplanted with Su9T01 cells via tail vein injection, and mouse survival was monitored. The HCQ-treated group showed a remarkably extended lifespan compared with the control group (Fig 3G). Collectively, these results suggest that CQ/HCQ generates a vigorous antileukemic effect through ATLL cell apoptosis.

## Discussion

In this study, we provided evidence that CQ and HCQ inhibit ATLL cell growth both in assays performed *in vitro* and in an *in vivo* mouse model of ATLL. ATLL cells exhibited high autophagic flux, and autophagy inhibition by CQ or HCQ promoted p47 protein recovery and inhibition of NF-κB activation, leading to apoptosis in ATLL cells.

Recently, we demonstrated that p47 degradation by the autophagic process enhances CADM1 expression in ATLL cells [13]. Interestingly, enforced expression of p47 in ATLL cells produced significant cell growth inhibition and downregulation of CADM1 [13]. These findings suggested that the inhibition of autophagy might be a potentially useful approach for ATLL treatment. Therefore, in the present study, we investigated the molecular effects of CQ and HCQ, which are recently described autophagy inhibitors [20]. We found that CQ/HCQ monotherapy inhibited the autophagic process, thus restoring p47 expression and inhibiting

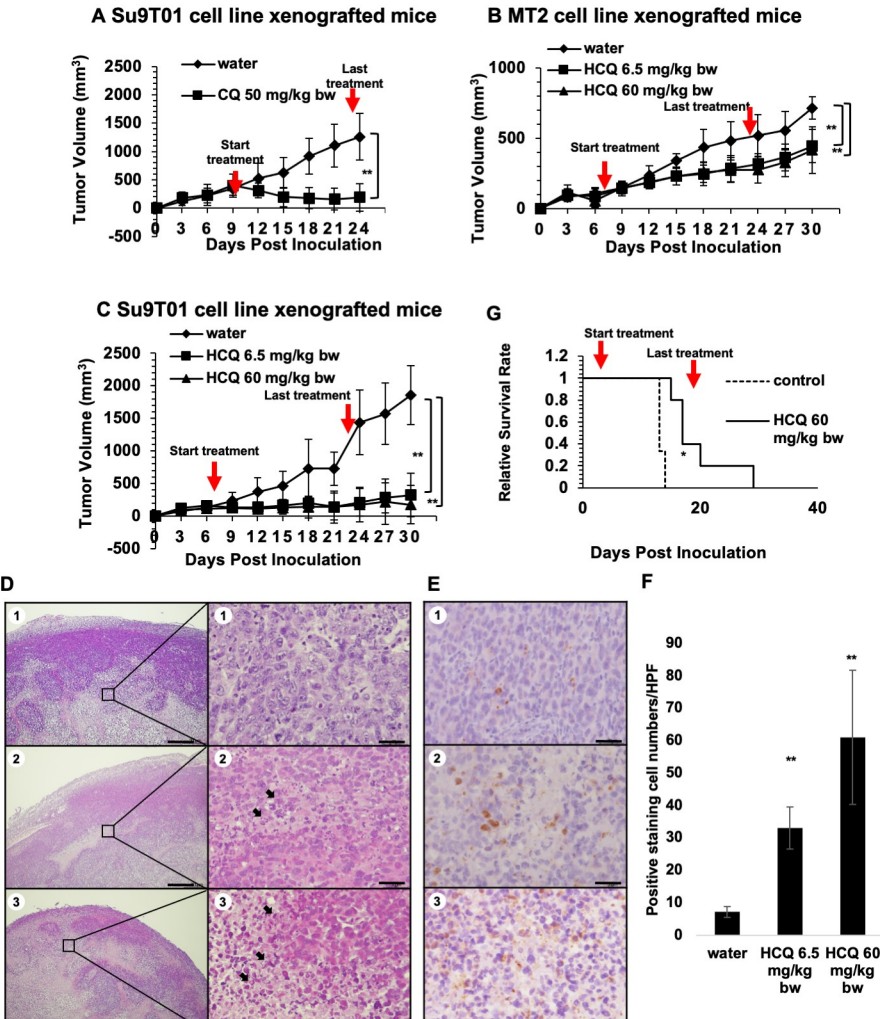

**Fig 3. CQ/HCQ inhibited tumor growth and extended the lifespan of ATLL cells-xenografted mice.** (**A**) Tumor volume of Su9T01 xenograft tumors in NOG mice intraperitoneally administered water (control) or 50 mg/kg bw CQ daily for 16 days. (**B, C**) Tumor volume of MT2 (**B**) or Su9T01 (**C**) xenograft tumors in NOG mice orally administered water (control), 6.5 mg/kg bw HCQ, or 60 mg/kg bw HCQ daily for 16 days. (**D, E**) Representative H&E (**D**) and IHC images of Caspase-3 (**E**) in tumor tissues from NOG mice orally administered water (**1**), 6.5 mg/kg bw HCQ (**2**), or 60 mg/kg bw HCQ (**3**) daily for 16 days. Scale bars: 300 μm (**D**), 30 μm (**E**). Higher magnification images of the boxed areas are shown on the right (Scale bars: 30 μm). The arrows indicate dead tumor cells. (**F**) Quantification Caspase3 stained cells is presented as the mean ± SD (n = 10 HPF, representative image was shown at Fig 3E). (**G**) Kaplan-Meier survival curves of the intravenous Su9T01 leukemia model orally treated with HCQ or water daily for 16 days. *p < 0.05, **p<0.01 compared to the control.

the NF-κB pathway with downregulation of CADM1 expression and that this NF-κB pathway inhibition ultimately led to caspase-3-related apoptosis induction.

CQ and HCQ, well-known antimalarial drugs, have been shown to inhibit the autophagic process and are recognized as new promising agents in cancer therapy. In *in vitro* experiments or *in vivo* mouse models, CQ/HCQ treatment was shown to induce apoptosis in solid cancer cells including melanoma [24] and pancreatic adenocarcinoma [25] cells, primary effusion lymphoma cells [26], and hematopoietic cancer cells, including acute myeloid leukemia (AML) cells [27, 28]. We observed that CQ/HCQ effectively inhibited ATLL cell proliferation and induced caspase-related apoptosis in a dose- and time-dependent manner. Moreover, our

*in vivo* study demonstrated significant tumor growth inhibition, a high number of apoptotic cells in tumor xenografts, and survival rate improvement by treatment with CQ/HCQ.

Autophagy is a multistage degradation process and can be inhibited at any autophagic flux stage; however, principally, autophagy inhibitors are divided into those acting on the early stage (initiation stage, such as PI3K inhibitors) and those acting on the late stage (fusion and degradation stage, such as autophagolysosome fusion, protease, and lysosomal inhibitors) [29, 30]. Interestingly, CQ and HCQ were initially considered lysosomal inhibitors, with the same mechanism as the late-stage autophagy inhibitor bafilomycin, but recently, it was revealed that CQ inhibits autophagolysosome fusion instead of impairing the lysosomal degradation capacity [20]. In line with these findings, we found that CQ effectively inhibited autophagy, which was accompanied by downregulation of NEMO, an NF-κB modulator that is degraded through the lysosomal pathway. In addition, CQ/HCQ treatment also prevented IκBα degradation, consistent with a previous report showing that the inhibition of autophagy by CQ blocks bortezomib-induced NF-κB activation in diffuse large B-cell lymphoma (DLBCL) cells [31]. Thus, CQ and HCQ may represent a new class of NF-κB inhibitors that affect the stability of important NF-κB signaling molecules, including p47 and IκBα, through autophagy inhibition.

CQ, together with HCQ, a less toxic derivative of CQ, is also useful in the treatment of autoimmune diseases, such as rheumatoid arthritis and systemic lupus erythematosus (SLE) [32]. Clinically relevant doses of HCQ selectively induced ATLL cell death in the ATLL xenograft mouse model, and no signs of adverse effects were observed. Approximately 50% of ATLL patients exhibit skin manifestations, including nodules, plaques, ulcers, erythroderma and purpura. Although these manifestations are often treated with various skin-directed therapies, such as phototherapy and radiation therapy, the overall survival of eruption-bearing patients is poorer than that of ATLL patients without an eruption [33, 34]. Therefore, CQ and HCQ could be promising drugs for ATLL therapy, which offers innovative insight for drug repositioning.

To sum up, we propose an anti-tumor model of CQ/HCQ in ATLL cells in Fig 4. CQ/HCQ treatment inhibited the high autophagic flux of ATLL cells. At this point, CQ/HCQ rescues

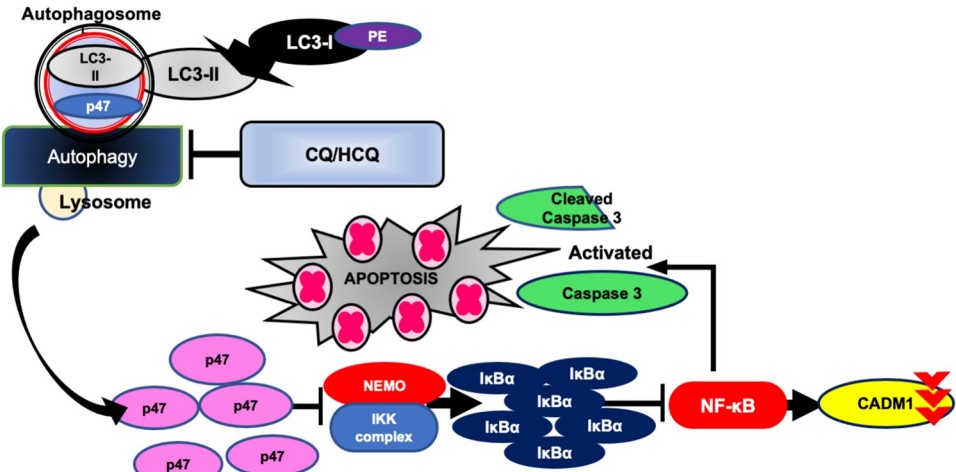

**Fig 4. Model illustrating the mechanism of CQ/HCQ-induced ATLL cell apoptosis by autophagy and NF-κB pathway inhibition.** In ATLL cells, p47 is degraded by the autophagy-lysosome pathway, leading to increased NEMO protein level and activation of IKK, which result in the constitutive activation of the NF-κB pathways. CQ and HCQ inhibit autophagy and increase p47 protein level, which inhibit NF-κB pathway activation along with loss of CADM1 expression, leading to caspase-3 related apoptosis.

p47 protein from the autophagy-lysosomal degradation pathway. The abundant p47 inhibited NEMO-IKK complex interaction, leading to downregulation of NEMO and preventing IKba degradation. Finally, the inhibition of NF-kB activation induced downregulation of CADM1 and caspase-related apoptosis.

In conclusion, this study showed that autophagy has an essential function in ATLL proliferation and that autophagy inhibition by CQ/HCQ might be a promising therapeutic strategy. Our study provides the rationale for a clinical trial evaluating HCQ in the future.

## Supporting information

**S1 Fig. CQ and HCQ treatment inhibit the growth of ATLL and HTLV-1-infected T-cell lines derived from HAM/TSP patients *in vitro*.** (**A**) Four ATLL cell lines (Su9T01, KK1, S1T, and MT2), two T-ALL cell lines (MOLT4 and JURKAT), and PBMCs from a healthy donor were treated with 50 μM CQ or 25 μM HCQ for 48 hours. Cell viability was determined by trypan blue. Relative cell viability was calculated based on the percentage of untreated cells (0 μM). Data are presented as the mean ± SD (n = 3). *p < 0.05 for Su9T01, KK1, S1T, MT2, MOLT4, or JURKAT cells compared to healthy PBMCs, respectively. (**B**) HCT1, HCT4, and HCT5 cell lines were treated with varying concentrations of CQ or HCQ for 48 hours. The relative cell viability was calculated as the percentage of untreated cells (0 μM). Data are presented as the mean ± SD (n = 3). *, #, or + P < 0.05, HCT1, HCT4, or HCT5 compared to Healthy PBMC, respectively.
(TIF)

**S2 Fig. ATLL cells exhibit high autophagic flux levels and CQ/HCQ inhibit autophagy and canonical NF-κB pathway.** (**A**) Su9T01, KOB, KK1 and S1T cell lines were treated with varying concentrations of CQ for 24 hours. The expression of LC-3 was examined by Western blot analysis. (**B**) Quantification of LC3 puncta/cell is presented as the mean ± SD (n = 3, representative image was shown at Fig 2A). (**C**) Western blot analysis of CADM1, p47, and the indicated NF-κB and autophagy signaling proteins was performed in KK1 cell lines after treatment with either 50 μM CQ or 25 μM HCQ for 6, 12, 18, 24 hours. β-actin was used as a loading control. The cropped gels/blots are used in the figure, and the full-length gels/blots are presented in S1 Raw images. (**D**) LC3-II turnover are presented as the mean ± SD (n = 2 independent experiments, representative blot was shown at S2B Fig). *p < 0.05, **p<0.01 compared to control.
(TIF)

**S3 Fig. CQ/HCQ inhibit tumor growth in ATLL cell xenograft mice.** Photograph of a tumor isolated from mice bearing the indicated cell lines after treatment with CQ, HCQ, or water.
(TIF)

**S4 Fig. CQ/HCQ treatment does not affect the mice's body weight.** Records of weight variations of mice at every 3 days. Arrow indicates the time of treatment.
(TIF)

**S1 File.**
(PDF)

**S1 Raw images.**
(PDF)

## Acknowledgments

The authors are grateful to Dr. Yasuaki Yamada (Nagasaki University, Japan) and Dr. Naomi-chi Arima (Kagoshima University, Japan) for providing cell lines.

## Author Contributions

**Conceptualization:** Yanuar Rahmat Fauzi, Kazuhiro Morishita.

**Formal analysis:** Yanuar Rahmat Fauzi.

**Funding acquisition:** Kazuhiro Morishita.

**Investigation:** Yanuar Rahmat Fauzi, Phawut Nueangphuet.

**Methodology:** Yanuar Rahmat Fauzi, Shingo Nakahata, Syahrul Chilmi, Tomonaga Ichikawa, Kazuhiro Morishita.

**Project administration:** Shingo Nakahata, Kazuhiro Morishita.

**Resources:** Syahrul Chilmi, Tomonaga Ichikawa, Ryoji Yamaguchi, Tatsufumi Nakamura, Kazuya Shimoda.

**Supervision:** Shingo Nakahata, Tatsufumi Nakamura, Kazuya Shimoda, Kazuhiro Morishita.

**Validation:** Yanuar Rahmat Fauzi, Ryoji Yamaguchi.

**Visualization:** Yanuar Rahmat Fauzi.

**Writing – original draft:** Yanuar Rahmat Fauzi.

**Writing – review & editing:** Yanuar Rahmat Fauzi, Shingo Nakahata, Kazuhiro Morishita.

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
