## [Decision Letter · Decision Letter 0]

15 Jun 2021

PONE-D-21-15148

Antitumor effects of chloroquine/hydroxychloroquine mediated by inhibition of the NF-κB signaling pathway through abrogation of autophagic p47 degradation in adult T-cell leukemia/lymphoma cells

PLOS ONE

Dear Dr. Morishita,

Thank you for submitting your manuscript to PLOS ONE. After careful consideration, we feel that it has merit but does not fully meet PLOS ONE’s publication criteria as it currently stands. Therefore, we invite you to submit a revised version of the manuscript that addresses in full the points raised during the review process including a careful revision of the references of the Bibliography section

We look forward to receiving your revised manuscript.

Kind regards,

Guillermo Velasco, Ph.D

Academic Editor

PLOS ONE

Journal Requirements:

"The authors are grateful to Dr. Yasuaki Yamada (Nagasaki University, Japan) and Dr. Naomichi Arima (Kagoshima University, Japan) for providing cell lines. This work was supported in part by Grants-in-Aid for Scientific Research (B) (25293081 and 17H03581) (KM) from the Japan Society for the Promotion of Science (JSPS) and by the Takeda Science Foundation (KM)."

"This work was supported in part by Grants-in-Aid for Scientific Research (B) (25293081 and 17H03581) (KM) from the Japan Society for the Promotion of Science (JSPS) and by the Takeda Science Foundation (KM)."

6. Please cite Supporting Information Figure S5 in your main manuscript.

Reviewers' comments:

Reviewer's Responses to Questions

**Comments to the Author**

1. Is the manuscript technically sound, and do the data support the conclusions?

Reviewer #1: Yes

Reviewer #2: Yes

2. Has the statistical analysis been performed appropriately and rigorously? 

Reviewer #1: Yes

Reviewer #2: Yes

3. Have the authors made all data underlying the findings in their manuscript fully available?

Reviewer #1: Yes

Reviewer #2: Yes

4. Is the manuscript presented in an intelligible fashion and written in standard English?

Reviewer #1: Yes

Reviewer #2: Yes

5. Review Comments to the Author

Reviewer #1: 1. The authors should replace most of the references to the proper ones in the introduction section. I attached revised ref lists for their conveniences.

2. No description of the PBL used in Fig.1A&B. in the Methods section. They should describe how they collected from healthy volunteers.

3. In clinical use, HCQ is less toxic than CQ. In Fig1A&B however, PBL viability seems dropping with less amount of HCG than CQ. Please rationalize this discordance.

4. Cmax of CQ is around 500nM. The viability of PBL and ATL cells look quite similar at this concentration. How the authors improve the efficacy of CQ/HCQ in clinical practice? It seems difficult to have sufficient efficacy with single drug treatment.

Reviewer #2: The study presented by Yanuar Rahmat Fauzi et al. entitled “Antitumor effects of chloroquine/hydroxychloroquine mediated by inhibition of the NF-κB signaling pathway through abrogation of autophagic p47 degradation in adult T-cell leukemia/lymphoma cells” shows that CQ and HCQ inhibit ATLL cell growth in in vitro assays as well as in a mouse model for ATLL. The autophagy inhibition mediated by CQ or HCQ promoted p47 protein recovery and inhibition of NF-κB activation, leading to apoptosis in ATLL cell lines.

The manuscript presents interesting data, it is novel, and conclusions are supported by the experimental data. However, the issues listed below should be addressed.

Major Points:

- Page 9. Authors claim that CQ and HCQ impair ATLL cell growth. However, they perform a MTT assay that assays metabolically active cells, then MTT is an indirect parameter of viability instead of cell growth. There are many situations where cells might not divide and be metabolically active. A measurement with trypan blue, or a supravital stain is recommended.

- Figure 2. Treatment with different CQ concentration and LC3 western blot for ST1 cell line should be depicted.

- The authors state that CQ treatment causes accumulation of autophagosomes. They support that state by the increased level of LC3-II in all ATLL cell lines in WB (figure S2A). It would be nice to show the increased LC3 positive dots by immunofluorescence.

- Page 11. Text says that the number of apoptotic cells in tumor xenografts was significantly increased by treatment with HCQ. However, authors do not show any statistical analysis to support such a thing. It would be necessary to quantify the caspase 3 positive cells or alternatively use another methodology such as TUNEL assay in xenografts and apply the appropriate statistical analysis.

- The authors used a Su9To1 cell injection model in the tail vein of mice, then survival was monitored. The authors must clarify this model better in Methods section. Additionally, they should give an explanation that justify the use of this model and/or insert a citation where the model is described.

Minor Points:

- Clarify in the text that autophagy refers to macroautophagy.

- In the title “CQ and HCQ inhibited autophagy and the canonical NF-κB pathway” it must be disclosed that it is in ATLL cells lines, so, it should be “CQ and HCQ inhibited autophagy and the canonical NF-κB pathway in ATLL cells lines”.

- Figure S2C, authors should clarify if LC3-II turnover calculation is from S2B WB.

- To enhance an easy reading, the results from figure S1 B and C must be included in Figure 1.

- How do authors explain the behavior of HTLV-1-infected T cell lines derived from HAM-TSP patients (HCT1, HCT4 and HCT5) in figure S1A?

- Figure S3. Please, clarify which cell line corresponds to each picture.

- In Figure 3, authors have to clarify images’ labels.

- In the title "CQ and HCQ inhibit tumor growth and extend the lifespan of ATLL cells in xenograft mice", Do CQ and HCQ extend the lifespan of ATLL cells or the mice?

- It would be nice to transfer to the discussion section the proposed model in Figure S5.

6. PLOS authors have the option to publish the peer review history of their article (what does this mean?). If published, this will include your full peer review and any attached files.

Reviewer #1: No

Reviewer #2: **Yes: **Maria Noé Garcia

---

## [Author Response · Author response to Decision Letter 0]

30 Jun 2021

Response to Reviewers:

June 24th, 2021

Dr. Guillermo Velasco

Academic Editor at PLOS ONE

Reference: Manuscript Number: PONE-D-21-15148

“Antitumor effects of chloroquine/hydroxychloroquine mediated by inhibition of the NF-κB signaling pathway through abrogation of autophagic p47 degradation in adult T-cell leukemia/lymphoma cells”

Dear Dr. Guillermo Velasco: 

We have revised the manuscript according to the editorial requests. We believe that we have answered all points raised during the review process and hope that the revised manuscript is now acceptable for publication.

Yours sincerely, 

Kazuhiro Morishita MD., PhD 

Director, HTLV-1/ATL Research, Education and Medical Facility, Faculty of Medicine, University of Miyazaki

Professor, Project for Advanced Medical Research and Development, 

Project Research Division, Frontier Science Research Center, University of Miyazaki, 

5200 Kihara, Kiyotake, Miyazaki, Miyazaki, JAPAN 889-1692

Phone: +81-985-85-9610 Fax: +81-985-85-0609

E-mail: kmorishi@med.miyazaki-u.ac.jp

Editor’s notes:

Journal Requirements:

In response to the editor’s comment, we have changed the writing style to follow PLOS ONE's style requirements. Please kindly let us know if we have missed something.

"The authors are grateful to Dr. Yasuaki Yamada (Nagasaki University, Japan) and Dr. Naomichi Arima (Kagoshima University, Japan) for providing cell lines. This work was supported in part by Grants-in-Aid for Scientific Research (B) (25293081 and 17H03581) (KM) from the Japan Society for the Promotion of Science (JSPS) and by the Takeda Science Foundation (KM)."

"This work was supported in part by Grants-in-Aid for Scientific Research (B) (25293081 and 17H03581) (KM) from the Japan Society for the Promotion of Science (JSPS) and by the Takeda Science Foundation (KM)."

In response to the editor’s comment, we have corrected the sentences, “"The authors are grateful to Dr. Yasuaki Yamada (Nagasaki University, Japan) and Dr. Naomichi Arima (Kagoshima University, Japan) for providing cell lines” in the Acknowledge section (page 16)

In response to the editor’s comment, we will provide the repository information.

In response to the editor’s comment, we have created a separate file, named S1_raw_images.pdf 

In response to the editor’s comment, we have authenticated the corresponding author’s ORCID iD

6. Please cite Supporting Information Figure S5 in your main manuscript.

In response to the reviewer’s comment, we moved Figure S5 to Figure 4 and added paragraph to the Discussion section 

Reviewer #1: 

1. The authors should replace most of the references to the proper ones in the introduction section. I attached revised ref lists for their conveniences.

In response to the reviewer’s comment, we have replaced the suggested references (References number 3, 4, 5, 7, 8, 9, and 12).

2. No description of the PBL used in Fig.1A&B. in the Methods section. They should describe how they collected from healthy volunteers.

In response to the reviewer’s comment, we added the sentences “Informed consent was obtained from all the patients and the healthy donor” in the Methods-Patient samples section (page 7).

3. In clinical use, HCQ is less toxic than CQ. In Fig1A&B however, PBL viability seems dropping with less amount of HCQ than CQ. Please rationalize this discordance.

Our study showed that IC50 of HCQ was lower than CQ. Consistent results with other cells (ATLL, T-ALL and HAM-TSP cells), indicating that HCQ is more potent than CQ. So, to reach a similar effect, HCQ needs a lower dose than CQ. Regarding toxicity, our in vivo studies using a recommended maintenance dose of HCQ for SLE in humans, and we did not observe any particular side effects during experiments.

4. Cmax of CQ is around 500nM. The viability of PBL and ATL cells look quite similar at this concentration. How the authors improve the efficacy of CQ/HCQ in clinical practice? It seems difficult to have sufficient efficacy with single drug treatment.

We understand that at 500nM our in vitro data does not show the desired effect. However, our in vivo study used a clinical maintenance recommended dose for SLE, and it showed a significant tumor growth inhibition and extended the survival. Currently, we do not have any data to explain this in vitro-in vivo discrepancy.

Reviewer #2: The study presented by Yanuar Rahmat Fauzi et al. entitled “Antitumor effects of chloroquine/hydroxychloroquine mediated by inhibition of the NF-κB signaling pathway through abrogation of autophagic p47 degradation in adult T-cell leukemia/lymphoma cells” shows that CQ and HCQ inhibit ATLL cell growth in in vitro assays as well as in a mouse model for ATLL. The autophagy inhibition mediated by CQ or HCQ promoted p47 protein recovery and inhibition of NF-κB activation, leading to apoptosis in ATLL cell lines.

The manuscript presents interesting data, it is novel, and conclusions are supported by the experimental data. However, the issues listed below should be addressed.

Major Points:

1. Page 9. Authors claim that CQ and HCQ impair ATLL cell growth. However, they perform a MTT assay that assays metabolically active cells, then MTT is an indirect parameter of viability instead of cell growth. There are many situations where cells might not divide and be metabolically active. A measurement with trypan blue, or a supravital stain is recommended.

In response to the reviewer’s comment, we performed a growth assay measured by trypan blue. The results showed that there was significance difference cell growth between KK1, S1T, MT2, MOLT4 and JURKAT compared to healthy PBMC (Fig S1 B). We added citation, sentences and figures in the Result and Supplementary methods section (page 9, Figure S1B)

2. Figure 2. Treatment with different CQ concentration and LC3 western blot for ST1 cell line should be depicted.

We think the reviewer mistyping S1T as ST1, as the correlated cell line in the figure 2 is S1T.

In response to the reviewer’s comment, we assayed the LC3 turnover of S1T that were either untreated or treated with various doses of CQ. The results showed increased levels of LC3-II in a dose dependent pattern (Fig S2A). We added sentences and figures in the Result section (page 11, Figure S1B)

3. The authors state that CQ treatment causes accumulation of autophagosomes. They support that state by the increased level of LC3-II in all ATLL cell lines in WB (figure S2A). It would be nice to show the increased LC3 positive dots by immunofluorescence.

In response to the reviewer’s comment, we performed an immunofluorescence assay and quantified the puncta. The results showed that CQ/HCQ treatment significantly increased the LC3 expression. We added sentences and figures in the Result section (page 11, Figure 2C and S2D)

4. Page 11. Text says that the number of apoptotic cells in tumor xenografts was significantly increased by treatment with HCQ. However, authors do not show any statistical analysis to support such a thing. It would be necessary to quantify the caspase 3 positive cells or alternatively use another methodology such as TUNEL assay in xenografts and apply the appropriate statistical analysis.

In response to the reviewer’s comment, we quantified the caspase 3 stained cells and analyzed the data. The results showed that the number of caspase 3 stained cells in tumor xenografts was significantly increased by treatment with HCQ. We added the sentences and figures in the results section (page 12, Figure 3G)

5. The authors used a Su9To1 cell injection model in the tail vein of mice, then survival was monitored. The authors must clarify this model better in Methods section. Additionally, they should give an explanation that justify the use of this model and/or insert a citation where the model is described.

In response to the reviewer’s comment, we added sentences, “After 3 days, the mice were randomly divided into 2 groups (water or 60 mg/kg bw/day HCQ, p.o.), and treatment was started immediately. HCQ or a control vehicle (water) was orally given for a 16-day period”, and insert the citation (page 9)

Minor Points:

1. Clarify in the text that autophagy refers to macroautophagy.

As the reviewer noted, we added the sentences “An important mechanism of the activated NF-κB signaling pathway in ATLL is the activation of the macroautophagy(herafter referred to as autophagy in the remainder of this manuscript)-lysosomal degradation of p47 (NSFL1C), a negative regulator of the NF-κB pathway” in the Abstract section (Page 3). 

2. In the title “CQ and HCQ inhibited autophagy and the canonical NF-κB pathway” it must be disclosed that it is in ATLL cells lines, so, it should be “CQ and HCQ inhibited autophagy and the canonical NF-κB pathway in ATLL cells lines”.

As the reviewer noted, we corrected the title “CQ and HCQ inhibited autophagy and the canonical NF-κB pathway” to “CQ and HCQ inhibited autophagy and the canonical NF-κB pathway in ATLL cells lines” in the Results section (page 10) .

3. Figure S2C, authors should clarify if LC3-II turnover calculation is from S2B WB.

As the reviewer noted, we changed the sentences “LC3-II turnover is presented as the mean ± SD (n = 2). *p < 0.05, **p<0.01 compared to the control” to “LC3-II turnover is presented as the mean ± SD (n = 2 independent experiments, representative blot was shown at Fig 2 A). *p < 0.05, **p<0.01 compared to the control” in the figure legend

4. To enhance an easy reading, the results from figure S1 B and C must be included in Figure 1.

As the reviewer noted, we moved the “Figure S1 B and C” to the “Figure 1 C and D.

5. How do authors explain the behavior of HTLV-1-infected T cell lines derived from HAM-TSP patients (HCT1, HCT4 and HCT5) in figure S1A?

As the reviewer noted, we edited the sentences “Interestingly, CQ or HCQ treatment also inhibited cell growth in HTLV-1-infected T cell lines derived from HTLV-1-associated myelopathy (HAM)/tropical spastic paraparesis (TSP) patients (Fig S1A)” to “Interestingly, CQ or HCQ treatment also inhibited cell growth in HTLV-1-infected T cell lines derived from HTLV-1-associated myelopathy (HAM)/tropical spastic paraparesis (TSP) patients in a dose-dependent manner (IC50 CQ = 34.73 +/- 24.89 and IC50 HCQ = 30.37 +/- 12.46) (Fig S1A)” in the Results section (page 9).

6. Figure S3. Please, clarify which cell line corresponds to each picture.

As the reviewer noted, we added the label to each group 

(A. Su9T1 cell line xenografted mice, B. MT2 cell line xenografted mice and C. Su9T1 cell line xenografted mice)

7. In Figure 3, authors have to clarify images’ labels.

As the reviewer noted, we added the label to each group 

(A. Su9T01 cell line xenografted mice, B. MT2 cell line xenografted mice and C. Su9T01 cell line xenografted mice)

8. In the title "CQ and HCQ inhibit tumor growth and extend the lifespan of ATLL cells in xenograft mice", Do CQ and HCQ extend the lifespan of ATLL cells or the mice?

As the reviewer noted, we corrected the title "CQ and HCQ inhibit tumor growth and extend the lifespan of ATLL cells in xenograft mice" to “CQ and HCQ inhibit tumor growth and extend the lifespan of ATLL cells xenografted mice” in the Results Section (page 11)

9. It would be nice to transfer to the discussion section the proposed model in Figure S5.

As the reviewer noted, we moved the proposed model in figure S5 to main figure 4, and added the paragraph “To sum up, we propose an anti-tumor model of CQ/HCQ in ATLL cells in Fig 4. CQ/HCQ treatment inhibited the high autophagic flux of ATLL cells. At this point, CQ/HCQ rescues p47 protein from the autophagy-lysosomal degradation pathway. The abundant p47 inhibited NEMO-IKK complex interaction, leading to downregulation of NEMO and preventing IKba degradation. Finally, the inhibition of NF-kB activation induced downregulation of CADM1 and caspase-related apoptosis” in the Discussion section (page 14)

---

## [Decision Letter · Decision Letter 1]

21 Jul 2021

PONE-D-21-15148R1

Antitumor effects of chloroquine/hydroxychloroquine mediated by inhibition of the NF-κB signaling pathway through abrogation of autophagic p47 degradation in adult T-cell leukemia/lymphoma cells

PLOS ONE

Dear Dr. Morishita,

Thank you for submitting your manuscript to PLOS ONE. After careful consideration, we feel that it has merit but does not fully meet PLOS ONE’s publication criteria as it currently stands. Therefore, we invite you to submit a revised version of the manuscript that addresses the points raised during the review process. Specifically please address the modifications indicated by reviewer 2.

We look forward to receiving your revised manuscript.

Kind regards,

Guillermo Velasco, Ph.D

Academic Editor

PLOS ONE

Journal Requirements:

Reviewers' comments:

Reviewer's Responses to Questions

**Comments to the Author**

1. If the authors have adequately addressed your comments raised in a previous round of review and you feel that this manuscript is now acceptable for publication, you may indicate that here to bypass the “Comments to the Author” section, enter your conflict of interest statement in the “Confidential to Editor” section, and submit your "Accept" recommendation.

Reviewer #1: All comments have been addressed

Reviewer #2: (No Response)

2. Is the manuscript technically sound, and do the data support the conclusions?

Reviewer #1: (No Response)

Reviewer #2: Yes

3. Has the statistical analysis been performed appropriately and rigorously? 

Reviewer #1: (No Response)

Reviewer #2: Yes

4. Have the authors made all data underlying the findings in their manuscript fully available?

Reviewer #1: (No Response)

Reviewer #2: Yes

5. Is the manuscript presented in an intelligible fashion and written in standard English?

Reviewer #1: (No Response)

Reviewer #2: Yes

6. Review Comments to the Author

Reviewer #1: They nicely reorganized their first manuscript to the revised one including some additional data and texts.

Reviewer #2: The revised manuscript by Yanuar Rahmat Fauzi et al. has been improved and authors have responded to my questions and concerns. However, I still see necessary to clarify the following three points:

1. Please, explain in the materials and methods section, how LC3 positive dots were quantified in immunofluorescence (Figure S2D), and mainly the used criteria.

2. On the same line, with the previous point, please explain in the materials and methods section, quantification criteria for caspase 3 in Figure 3G.

3. The quality of figures 3 D and E should be improved.

7. PLOS authors have the option to publish the peer review history of their article (what does this mean?). If published, this will include your full peer review and any attached files.

Reviewer #1: No

Reviewer #2: **Yes: **Maria Noé Garcia

---

## [Author Response · Author response to Decision Letter 1]

28 Jul 2021

Response to Reviewers:

July 22nd, 2021

Dr. Guillermo Velasco

Academic Editor at PLOS ONE

Reference: Manuscript Number: PONE-D-21-15148R1

“Antitumor effects of chloroquine/hydroxychloroquine mediated by inhibition of the NF-κB signaling pathway through abrogation of autophagic p47 degradation in adult T-cell leukemia/lymphoma cells”

Dear Dr. Guillermo Velasco: 

We have revised the manuscript according to the editorial requests. We believe that we have answered all points raised during the review process and hope that the revised manuscript is now acceptable for publication.

Yours sincerely, 

Kazuhiro Morishita MD., PhD 

Director, HTLV-1/ATL Research, Education and Medical Facility, Faculty of Medicine, University of Miyazaki

Professor, Project for Advanced Medical Research and Development, 

Project Research Division, Frontier Science Research Center, University of Miyazaki, 

5200 Kihara, Kiyotake, Miyazaki, Miyazaki, JAPAN 889-1692

Phone: +81-985-85-9610 Fax: +81-985-85-0609

E-mail: kmorishi@med.miyazaki-u.ac.jp

Editor’s notes:

Journal Requirements:

 

Reviewer #1: They nicely reorganized their first manuscript to the revised one including some additional data and texts.

Reviewer #2: The revised manuscript by Yanuar Rahmat Fauzi et al. has been improved and authors have responded to my questions and concerns. However, I still see necessary to clarify the following three points:

1. Please, explain in the materials and methods section, how LC3 positive dots were quantified in immunofluorescence (Figure S2D), and mainly the used criteria.

In response to the reviewer’s comment, we added the sentences “Average numbers of LC3 puncta/cell were quantified by blinded manual counting of puncta, with at least 30 cells counted per group.” in the Supplementary Methods section (page 6).

2. On the same line, with the previous point, please explain in the materials and methods section, quantification criteria for caspase 3 in Figure 3G.

In response to the reviewer’s comment, we added the sentences “Quantification of anti-cleaved caspase-3 positive cells was performed by blinded manual counting the number of positive cells/High Power Field (HPF).” in the Supplementary Methods section (page 7).

3. The quality of figures 3 D and E should be improved.

In response to the reviewer’s comment, we have changed figures 3 D and E.

---

## [Editor Report · Decision Letter 2]

4 Aug 2021

Antitumor effects of chloroquine/hydroxychloroquine mediated by inhibition of the NF-κB signaling pathway through abrogation of autophagic p47 degradation in adult T-cell leukemia/lymphoma cells

PONE-D-21-15148R2

Dear Dr. Morishita,

We’re pleased to inform you that your manuscript has been judged scientifically suitable for publication and will be formally accepted for publication once it meets all outstanding technical requirements.

Kind regards,

Guillermo Velasco, Ph.D

Academic Editor

PLOS ONE
---

## [Editor Report · Acceptance letter]

6 Aug 2021

PONE-D-21-15148R2 

Antitumor effects of chloroquine/hydroxychloroquine mediated by inhibition of the NF-κB signaling pathway through abrogation of autophagic p47 degradation in adult T-cell leukemia/lymphoma cells 

Dear Dr. Morishita:

I'm pleased to inform you that your manuscript has been deemed suitable for publication in PLOS ONE. Congratulations! Your manuscript is now with our production department. 

Kind regards, 

on behalf of

Dr Guillermo Velasco 

Academic Editor

PLOS ONE